# Speckle-Correlation Scattering Matrix Approaches for Imaging and Sensing through Turbidity

**DOI:** 10.3390/s20113147

**Published:** 2020-06-02

**Authors:** YoonSeok Baek, KyeoReh Lee, Jeonghun Oh, YongKeun Park

**Affiliations:** 1Department of Physics, Korea Advanced Institute of Science and Technology, Daejeon 34141, Korea; lovebaek@kaist.ac.kr (Y.B.); kyeo@kaist.ac.kr (K.L.); jhun18@kaist.ac.kr (J.O.); 2Tomocube Inc., Daejeon 34109, Korea

**Keywords:** lensless imaging, sensing, holography, speckle, scattering media, microscopy

## Abstract

The development of optical and computational techniques has enabled imaging without the need for traditional optical imaging systems. Modern lensless imaging techniques overcome several restrictions imposed by lenses, while preserving or even surpassing the capability of lens-based imaging. However, existing lensless methods often rely on a priori information about objects or imaging conditions. Thus, they are not ideal for general imaging purposes. The recent development of the speckle-correlation scattering matrix (SSM) techniques facilitates new opportunities for lensless imaging and sensing. In this review, we present the fundamentals of SSM methods and highlight recent implementations for holographic imaging, microscopy, optical mode demultiplexing, and quantification of the degree of the coherence of light. We conclude with a discussion of the potential of SSM and future research directions.

## 1. Introduction

Traditionally, lenses have played a critical role in imaging. They collect scattered or emitted light from an object and convert a diverging wavefront of light into a converging one so that propagating light forms an image at a distance. However, the use of these optical components imposes several limitations, including the need for complicated imaging systems and specific imaging conditions, optical aberration, or the lack of proper materials for certain wavelengths such as deep ultraviolet or X-ray.

An alternative approach to address these issues is lensless imaging, wherein the reversal of light propagation is achieved numerically rather than physically [1]. For example, lensless imaging based on holography [2] interprets a recorded image as a hologram, and its digital reconstruction yields an optical field image of an object [3,4]. In particular, in-line digital holography has been widely utilized for biomedical and biophysical applications [1,5,6,7]. However, they are effective only for weakly scattering objects, and thus have limited applicability. Alternatively, phase retrieval algorithms [8,9] can be used for lensless imaging that utilizes coherent light. These methods encode phase information into intensity based on propagation in free space [10,11] and optical fibres [12], diffraction from masks [13,14,15,16], or modulated illumination [17]. Iterative algorithms are then used to obtain holographic images by utilizing acquired intensity images and pre-defined conditions as constraints. Similarly, lensless imaging of incoherent light has been demonstrated using coded apertures [18,19,20,21,22], wavefront shaping [23], scattering layers [24,25,26,27], and structured illumination [28]. However, for these methods, a priori information about the imaging conditions and the sample properties is crucial to correct image reconstruction. Recently, deep learning has been implemented in lensless imaging using scattering layers and optical fibres [29,30,31,32,33,34,35].

Herein, we review a unique technique known as speckle-correlation scattering matrix (SSM) [36] that measures complex amplitudes of light by exploiting the ‘randomness’ of scattering media. The role of the scattering media is described by a transmission matrix (TM or **T**) [37,38,39,40,41] that expresses a relationship between the incident ***E*_insicent_** and transmitted fields ***E*_transmitted_** using a matrix equation as follows:(1)$$E_{\mathrm{transmitted}}=TE_{\mathrm{incident}}$$
where **T** is deterministic for a given optical configuration. Thus, for exact knowledge about TM, a scattering medium can be utilized for imaging and wavefront shaping [42,43,44,45,46,47]. However, TM-based techniques generally require interferometric systems to calibrate the TMs and transmitted fields [46,48,49], or they are limited to the imaging of the amplitude of light [50].

The SSM method is different from existing methods for two reasons. Firstly, this approach facilitates deterministic holographic imaging from a single intensity measurement. Secondly, it utilizes the universal statistical property of the speckle phenomenon without a priori information or volumetric data correction. Recent studies have shown that the SSM methods achieve holographic imaging and sensing through turbidity [51,52], which will pave the way for new applications in various spectral domains.

The key differences and advantages of the SSM method are illuminated in Figure 1, compared with conventional approaches. Intensity imaging or photography only measured time-averaged intensity information of light. Thus, wavefront or phase information of light cannot be recorded, thus volumetric and depth information cannot be retrieved (Figure 1a). The inaccessibility to phase information in conventional intensity imaging, or the so-called phase problem [53], is a classical problem in optics and photonics, and has resulted in the development of various alternative approaches in diverse fields, ranging from X-ray crystallography [54,55], holography [2,56], communication [56], tomographic image reconstruction [57], and three-dimensional display [58,59].

Holography or interferometry overcome the phase problem (Figure 1b). By interfering a sample field with a well-defined reference beam, the phase information of a sample is modulated on intensity patterns and retrieved. The use of holography has been widely exploited for various applications where phase information becomes important, including bioimaging and metrology [60,61,62]. However, it requires a complicated interferometric setup or a computationally heavy phase retrieval process.

Recently, wavefront shaping or sensing techniques have shown potential for imaging through turbidity (Figure 1c). Because multiple light scattering in turbid media is a deterministic process, and phase information plays an important role in focusing and delivering optical information through complex media [43,63,64,65,66]. Still, the measurements of speckle fields require interferometry, and it results in difficulties in translation into practical applications. The concept of the SSM is to deliver optical field information through a turbid layer with a simple setup (Figure 1d), by exploiting the physical characteristics of light scattering and mathematical properties of auto-correlation. In this review, we present the principle of SSM and summarize recent SSM-based techniques.

## 2. Principle of SSM

Let us consider a simple setup composed of an optical diffuser and a camera (Figure 2). Suppose there is an unknown input field x=∑i=1Nxiei, where ei is a basis that satisfies ∑i=1Neiei†=IN; xi is the complex-valued coefficient of the *i*-th basis; and *N* is the number of input optical modes, x∈ℂN. The basis can be arbitrary, but it is conventionally set to the position or reciprocal (Fourier) basis. The speckle field **y** at the camera plane can be expressed using the TM of the diffuser, y=Tx=∑i=1Nxiti, where ti=Tei corresponds to a speckle field for x=ei and the *i*-th column of TM. The speckle field is expressed using *M* complex values y∈ℂM, where *M* is the number of optical modes observed at the camera.

The captured speckle intensity image is the square modulus of the speckle field y∘y*, where ∘ represents an element-wise (or Hadamard) product. The SSM (Z∈ℂN×N) can be expressed in terms of the TM and the measured speckle pattern as follows: (2)Zij=1σiσj[〈ti*∘tj∘y*∘y〉−〈ti*∘tj〉〈y*∘y〉]
where 〈u〉=1n∑i=1nui for u∈ℂn, and σi=〈ti*∘ti〉 is a normalization factor. Exploiting the Gaussian statistics of the speckle fields, each speckle field can be regarded as a random vector that satisfies the Wick’s (or Isserlis’) theorem [68],
(3)〈ti*∘tj∘y*∘y〉=〈ti*∘y〉〈tj∘y*〉+〈ti*∘y*〉〈tj∘y〉+〈ti*∘tj〉〈y*∘y〉
which is the key idea of the SSM method. Applying Equation (3), Equation (2) can be reduced to the following:(4)Zij=1σiσj[〈ti*∘y〉〈tj∘y*〉+〈ti*∘y*〉〈tj∘y〉]  =1ti†titj†tj[(ti†y)(tj†y)*+(ti†y*)(tj†y*)*]
where n〈u*∘v〉=u†v for u, v∈ℂn. For a large oversampling ratio γ=MN, that is, γ≫1, the orthogonality relations between speckle fields hold, that is, ti†tj≈||ti||2δij and ti†tj*≈0, because of random phase matching. Finally, Equation (4) is reduced to the following:(5)Zij=xixj*
from which the complex-coefficients of the incident field x=(x1,x2,⋯,xN) can be retrieved by eigenvector analysis. The assumption we made in the last step is γ≫1. However, in practice, a significantly large γ is not preferred because this requires significant oversampling of the information. For this reason, γ is usually set to ~10 in experiments and an additional error reduction algorithm is employed (see Section 4 for the details of the oversampling ratio and the error reduction algorithm).

## 3. Implementations

### 3.1. Reference-Free Holographic Imaging

Holography has proven to be a powerful technique for the measurement of both the amplitude and phase of light. In particular, quantitative phase imaging facilitates label-free rapid imaging of live cells and tissues. This approach has been utilized in various biological and clinical applications [61]. Accurate quantification in holography is based on the interference between an unknown field and a well-defined reference field. Although interferometric methods are simple and powerful, the use of an external reference field imposes technical challenges [61]. Interferometry generally requires a coherent light source and a complicated highly stable setup that is not only vulnerable to noise, but also difficult to realize in practice, for example, in the X-ray regime.

Lee and Park proposed reference-free holographic imaging based on the SSM [36], which is differentiated by the general principle and deterministic nature of the approach. In the experiment, an optical diffuser that transforms the incident field into a speckle field was placed in front of an image sensor to perform holographic imaging (Figure 3a). They calibrated the TM of the diffuser by projecting the image of a spatial light modulator and a reference beam, followed by a systematic change of the phase of the spatial light modulator (SLM) (Figure 3b). The calibration is based on phase-shifting interferometry, where three different phase shifts are applied to each input mode. Additionally, the speckle pattern of the reference beam is measured. Thus, the total 3*N* + 1 images were used for *N* input modes. The calibration time took 20 min for 3960 input modes. After calibration of the TM, the reference beam was blocked, and an object was imaged. Using the calibrated TM and the speckle image of an object, the *Z* matrix in Equation (2) is reconstructed. Then, by performing a singular value decomposition of the *Z* matrix, the complex coefficients for input modes are obtained (Equation (5)). The obtained complex coefficient corresponds to the holographic image of the object. In the experimental demonstration, holographic images of amplitude-modulated and phase-modulated incident fields were successfully measured (Figure 3c). The holographic image obtained by scattering from two dices was also measured, demonstrating three-dimensional imaging capability via numerical propagation (Figure 3d).

Given that the SSM is based on the exact knowledge about a TM, it is advantageous to obtain a priori information instead of experimental calibration of the TM. For example, a designed scattering layer can be utilized to address the issue of long-term stability and measurement noise in the calibration. In this regard, Kwon et al. adopted a metasurface diffuser for holographic imaging based on the SSM [69]. The metasurface diffuser composed of amorphous silicon (α–Si) cuboids generates speckle fields similar to an optical diffuser (Figure 4a). Given that the TM of the metasurface diffuser is known in advance, holographic imaging is achieved by only measuring the speckle pattern. In the experiment, holographic imaging of amplitude targets was demonstrated with a metasurface diffuser (Figure 4b).

### 3.2. High-Resolution Long-Working-Distance Holographic Microscopy

The performance and the imaging condition of a conventional lens-based imaging system vary greatly depending on the optical components. For example, in optical microscopy, the resolution and the working distance are inversely related because the objective lens has a fixed focal length and a finite aperture. The trade-off between resolution and working-distance poses a significant challenge in imaging three-dimensional samples, as well as samples in bulky containers and devices. A straightforward solution to simultaneously achieve high-resolution and long working-distance-imaging is to enlarge the size of the aperture of the imaging lens. However, this is often impractical because lenses with large apertures suffer from severe optical aberration. Although there have been attempts to utilize customized objective lenses with large apertures [70,71], they are typically incompatible with microscopic platforms and introduce stability issues.

Baek et al. proposed a method to achieve high-resolution long-working-distance imaging by exploiting the SSM [51]. In the experiment, the detection part of a holographic microscope was replaced with a scattering layer (Figure 5a). First, the TM was calibrated by modulating incident waves using a digital micromirror device [72,73]. Speckle patterns were then generated by illuminating samples with a normally incident plane wave. Finally, amplitude and phase images of microscopic samples, including a resolution target and a human red blood cell, were measured by applying the SSM to the speckle images. This method results in flexible imaging conditions because the spatial resolution, field of view, and working distance can be easily changed by relocating the scattering layer. In addition, high-resolution long-working-distance imaging is facilitated by the scattering layer. This is possible because it is easy to fabricate a large scattering layer and this layer only needs to generate random speckle fields without considering optical aberration. In the experiment, high-resolution long-working-distance imaging was demonstrated with a scattering layer with a diameter of 26.5 mm. They obtained aberration-free holographic images with a numerical aperture (NA) of 0.7 and a working distance of 13 mm. This contrasts with severe optical aberrations in holographic images measured with a long-working-distance objective lens with the same NA and a much shorter working distance of 2 mm (Figure 5b).

### 3.3. Probing Polarization and Coherence of Light

The complete characterization of light must take coherence into account because light can be partially coherent in space, time, and polarization state. Experimentally, this requires the measurement of the amplitude, phase, and polarization state of the degree of coherent light, and its statistical distribution in space and time. For example, Stokes polarimetry [74] uses phase-shifting interferometry for every possible pair of coherent states. However, such an approach is not practical because of the enormous number of required measurements.

Lee and Park addressed this issue by generalizing the SSM to incoherent states of light [67]. The generalization was achieved using a birefringent diffuser, which generates spatially uncorrelated speckle patterns for different polarization states and different wavefronts of the incident light. For partially coherent or incoherent states of light, the birefringent diffuser generates a speckle pattern that is a linear summation of speckle patterns owing to the coherent states as follows:(6)$$y^{*}\circ y=\underset{\alpha }{\sum }P_{\alpha }y_{\alpha }^{*}\circ y_{\alpha }$$
where $y_{\alpha }$ is the speckle field generated by the *α*-th coherent state and $P_{\alpha }$ is the statistical weight. In such a case, the SSM can be expressed as a linear summation of SSMs for coherent states as follows:(7)$$Z_{ij}=\underset{\alpha }{\sum }\frac{P_{\alpha }}{\sigma_{i}\sigma_{j}}\left[\langle t_{i}^{*}\circ t_{j}\circ y_{\alpha }^{*}\circ y_{\alpha }\rangle -\langle t_{i}^{*}\circ t_{j}\rangle \langle y_{\alpha }^{*}\circ y_{\alpha }\rangle \right]$$

Following the same steps in Section 2, the SSM can be rewritten as follows:(8)$$Z_{ij}=\underset{\alpha }{\sum }P_{\alpha }x_{i,\alpha }x_{j,a}^{*}$$
where $x_{i,\alpha }$ is the complex coefficient of the *α*-th coherent state for the *i*-th basis $e_{i}$. The result shows that the SSM can be used to generate the amplitude and phase of coherent states along with their statistical weights from a single speckle image. These measurements represent the coherency matrix in classical optics, which is analogous to the density matrix in quantum optics.

The experimental demonstration was conducted using an unpolarized laser and a birefringent diffuser made of rutile nanoparticles. First, the birefringent TM was calibrated using an SLM similar to the method in [36]. Right circularly-polarized light generated by a polarized or unpolarized HeNe laser source was modulated using an SLM. As a result, the polarized light has a spatially varying polarization, whereas the unpolarized light remains unpolarized. The right circularly-polarized has a spatially varying phase delay. A single coherent speckle pattern is generated in the case of the polarized light. The SSM method is used to reconstruct holographic images of two orthogonal polarization states, facilitating the visualization of the spatially varying polarization (Figure 6a). For the unpolarized light, two incoherent speckle patterns are generated. From Equation (8), the statistical weights of the right and the left circularly-polarized states are measured. The experimental values are 0.52 and 0.48 for the right and left circularly polarized states, respectively, which is in good agreement with the expected value of 0.5. At the same time, holographic images of two polarization states were successfully measured (Figure 6b). Although the demonstration was conducted using two incoherent polarization states, the result suggests that the SSM can be used to characterize multiple incoherent states such as spatial partially coherent or incoherent light.

### 3.4. Orbital Angular Momentum Mode Demultiplexing

To meet the growing demands for greater data capacity in optical communication, orbital angular momentum (OAM) multiplexing has been proposed [75]. By encoding information into the OAM states of light, data transmission has been demonstrated in free-space [76,77] and optical fibres [78]. OAM multiplexing is expected to facilitate significant advances in data capacity because it allows an infinite number of eigenstates and is compatible with communication techniques such as wavelength-division and polarization-mode multiplexing, as well as various modulation schemes. However, OAM demultiplexing based on holograms is limited when the information of OAM states is scrambled by atmospheric scattering and mode-coupling in optical fibres.

Gong et al. have demonstrated OAM demultiplexing in a severe scattering environment based on the SSM [79]. In the experiment, OAM states generated by the digital micromirror device are scattered by the media (Figure 7a). The free-space propagation of the scattered OAM states generates a speckle pattern at the image sensor. By exploiting the SSM, Gong and colleagues successfully recovered the input OAM states from the speckle image. The estimated weights for OAM states exhibited good agreements with the correct values with an error rate of less than 10% for monochromatic light (Figure 7b). This is in contrast with holographic demultiplexing, wherein the demultiplexing fails even for small misalignment. Further experiments have demonstrated successful data transfer of monochromatic and colour images based on binary amplitude modulation of OAM states (Figure 7c). Finally, they measured the relative phase between OAM states, and showed benefits in coherent optical communication systems (Figure 7d). Although a strong scattering environment is assumed in [79], it is possible to extend the OAM demultiplexing to a weak scattering environment by placing scattering media near the detection region. To implement the demultiplexing method in a dynamic environment such as in the presence of atmospheric turbulence or the perturbations of an optical fibre, rapid calibration of the TM is necessary.

### 3.5. Measuring the Transmission Matrix of a Multi-Mode Fibre

A known TM of an optical system such as a scattering media or a multi-mode fibre allows for direct control and the detection of waves [39,43,63]. However, the calibration of the TM is largely based on interferometry, which involves the modulation of incident waves and the measurement of the interference between a scattered and a reference wave [37,48]. As such, interferometric methods require bulky setups and high-stability throughout the calibration procedure. As an alternative, common-path interferometry or phase retrieval algorithms can be used. Common-path interferometry utilizes an unmodulated speckle field as a reference [38,50,80]. Although this obviates the need for external reference beams, the calibrated TM is not perfect because of the unknown amplitude and phase of the speckle reference. Phase retrieval algorithms use pre-defined incident patterns and utilize intensity images of the transmitted waves to identify the TM [81,82,83,84,85]. However, they require a large number of measurements for precise calibration and the uniqueness of the iterative algorithm is unclear.

Huang et al. proposed a method to calibrate the TM based on the SSM and an iterated extended Kalman filter [52]. Conceptually, they switch the role of the input field and the TM of a conventional SSM. They define the modulation matrix **X**, which is the set of known input fields $x_{k}$ as row vectors,
(9)$$X={\left(x_{1},x_{2},\cdots x_{L}\right)}^{Τ}$$
where *L* is the number of different input modes that were attempted. We can then define the output matrix using the TM as follows:(10)$$Y=XT^{Τ}$$
where, similarly, $Y={\left(y_{1},y_{2,}\cdots ,y_{L}\right)}^{Τ}$ and $y_{k}=Tx_{k}$. If the modulation matrix X is set to follow a complex Gaussian distribution, the modified SSM can be applied to each column vector of Equation (10),
(11)$$Z_{ijk}^{\prime }=\frac{1}{\sigma_{i}\sigma_{j}}\left[\langle s_{i}^{*}\circ s_{j}\circ u_{k}^{*}\circ u_{k}\rangle -\langle s_{i}^{*}\circ s_{j}\rangle \langle u_{k}^{*}\circ u_{k}\rangle \right]$$
where $s_{i}$ and $u_{k}$ are the *i*- and *k*-th columns of **X** and **Y**, respectively, and $\sigma_{i}=s_{i}^{*}\circ s_{i}$. Following the steps in Section 2, singular value decomposition of Equation (11) results in the *k*-th column vector of $T^{Τ}$ (or the row vector of **T**).

Huang et al. refined the result of the SSM using an iterated extended Kalman filter, wherein the measurement error induced by Gaussian random noise is minimized via an iterative algorithm (Figure 8a). Numerical simulation revealed that *L* = 4*N* modulations for *N* input modes result in a correct TM from which ideal wavefront shaping performance is achieved (Figure 8b,c). In an experiment with a multi-mode fibre, the TM was successfully measured, and enhancement of the focal intensity of 286 (P1 in Figure 8c) with 520 input modes was achieved. Without the extended Kalman filter, the achieved enhancement was 76 (P1′ in Figure 8c). This work demonstrates that the SSM can be utilized in other linear systems to measure signals, other than holographic images.

## 4. Criteria for SSM

In this section, we discuss the conditions under which the SSM method can achieve the desired performance, and an iterative algorithm to improve the quality of reconstructed images.

### 4.1. Complex Gaussian Statistics of Speckle Fields

The key assumption in Equation (3) is the complex Gaussian distribution of the entities in a TM [57,86,87]. This condition can be tested using the dimensionless conductance *g* [88], defined as follows:(12)$$g\approx 2\pi A\lambda^{-2}\overset{¯}{T}$$
where *A* is the beam size that is incident on the diffusive or scattering layer, *λ* is the wavelength of the incident light, and $\overset{¯}{T}$ is the mean transmittance of the layer. It is known that the TM has uncorrelated random numbers when g≫M for sufficiently large *N* and *M* [88], which is indicative of a complex Gaussian distribution of the TM according to the central limit theorem. The condition g≫M can be satisfied by adjusting the beam size, the number of output modes, or using a different diffusive or scattering layer. For example, the works of [51,67] indicate *g* values of approximately 2 × 10^8^, which is much greater than their respective output modes of 5 × 10^5^ and 5 × 10^4^.

In addition to the complex Gaussian distribution of the TM, the transmitted field **y** must follow a complex Gaussian distribution. Although, in most cases, **y** follows a complex Gaussian distribution for an uncorrelated TM, there are exceptions wherein non-Gaussian speckle images are produced. Wavefront shaping and optical phase conjugation techniques have demonstrated that the transmitted field can yield a focus [43,64] or images [89,90], by appropriately controlling the incident fields. In these examples, **y** does not follow a complex Gaussian distribution and Equation (3) does not hold. Nevertheless, these exceptions do not limit the applicability of the SSM because random speckles can be easily generated by translating or tilting the scattering layers [90,91].

Finally, it is important to choose an appropriate basis for the SSM. This is because a change of the basis can dramatically change the statistics of the elements of the TM. For instance, the diagonalization of an uncorrelated TM results in a diagonal matrix wherein the columns do not follow complex Gaussian statistics. Conversely, it is also possible to construct a complex Gaussian TM even in the absence of a diffusive or scattering layer with a change of basis. Thus, the TM should be carefully analyzed for a given basis to fully exploit the advantages of the SSM.

### 4.2. Oversampling Ratio 

A large oversampling ratio is necessary to ensure the orthogonality between the speckle fields, otherwise Equation (4) cannot be reduced to Equation (5). In principle, the larger the oversampling ratio, the more accurate the SSM. The results of numerical simulations show that the SSM alone provides a correct solution when the oversampling ratio is much greater than 10 [36] (Figure 9). However, when incoherent input modes are considered, the required oversampling ratio linearly increases with the total number of incoherent modes [67]. In practice, a large oversampling ratio is difficult to achieve because an image sensor has a finite pixel count, and a reduction in the number of input modes negatively impacts the field of view or resolution. For this reason, an iterative algorithm is usually employed (see Section 4.3).

It should be emphasized that the effective number of output modes is determined by the number of speckles, not by the pixel count of an image sensor. Thus, optimizing the size of a speckle grain is crucial to maximize the oversampling ratio, and to improve the image quality using the SSM. Typically, adjusting the distance between a scattering layer and the image sensor or the size of the incident beam can control the size of a speckle grain. However, as the oversampling ratio increases, fewer pixels of an image sensor are allocated to a speckle. This could lead to noise in speckle images because of the limited signal-to-noise ratio of an image sensor.

### 4.3. Iterative Algorithms for Error Reduction

In practice, the limited signal-to-noise ratio of image sensors, the deviation of the TM from the complex Gaussian distribution, and the limited oversampling ratio lead to an error in the SSM. To address these issues, an iterative algorithm can be employed. In this section, the Gerschberg–Saxton type iterative algorithm in [36] is introduced. First, the retrieved field $x_{n}$ is multiplied by the TM to obtain the transmitted field $y_{n}=Tx_{n}$, where the subscript *n* is the number of iterations. The amplitude of $y_{n}$ is then replaced with the square root of the measured intensity |y| to yield an updated approximation of the transmitted field $y_{n+1}=\frac{\left|y\right|y_{n}}{\left|y_{n}\right|}$. By applying the pseudoinverse of the TM to $y_{n+1}$, an updated incident field $x_{n+1}$ is obtained. The entire process is repeated until no significant change of **x** is made between iterations. The algorithm quickly converges and provides the exact solution, even for a small oversampling ratio (γ=4) [36], without imposing assumption or regularization parameters. The numerical solution shows that the algorithm is slightly more effective for the phase component than the amplitude component of an image for the same SNR and the same oversampling ratio (Figure 9). The effectiveness of the iterative algorithm is maintained when multiple incoherent input modes are considered (Figure 10a–d) [67].

## 5. Conclusions and Outlook

The SSM method provides a mathematical framework to interpret random speckle images for imaging and sensing applications, wherein the intensity speckle can be regarded as an interferogram that contains the information of the input field. Recent studies have demonstrated the great potential of the SSM in imaging and sensing through scattering media. Although experimental demonstrations have focused on the utilization of visible light, the SSM method can be particularly useful when the choice of optical components is limited, such as in X-ray and Terahertz regimes. Furthermore, by utilizing the multispectral transmission matrix [93] and the time-resolved transmission matrix [94], new possibilities in multi-spectral imaging, spectroscopy, ultrafast imaging, and holographic memory are expected [95,96,97,98]. The SSM can also be extended to aperiodic structures [99], diffusive surfaces [100], and multi-mode fibres that generate spatially uncorrelated speckle fields. The use of these devices may facilitate applications in various optical configurations such as non-line-of-sight imaging, endoscopy, and optical communications.

For successful implementations, the role of the error reduction algorithm is important, because in practice, the accuracy of the SSM is limited by the imperfect orthogonality between the speckle fields and the finite oversampling ratio. Although studies have shown the efficacy of the Gerschberg–Saxton type algorithm, advanced algorithms are needed to account for the errors in the measurement and physical properties of scattering layers, such as the memory effect [91,101,102,103]. The SSM method is not necessarily the most efficient way to extract information from a single speckle image. We believe that more efficient methods will be developed in the future and facilitate real-time imaging applications. A non-linear approach, such as deep learning, is a viable candidate.

## Figures and Tables

**Figure 1 sensors-20-03147-f001:**
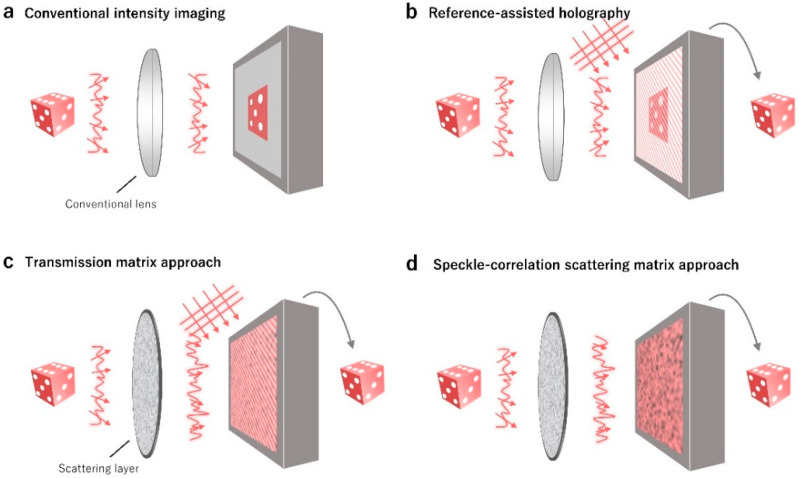
The comparison between the speckle correlation scattering matrix (SSM) method and conventional imaging methods. (**a**) Intensity imaging does not access the phase information of light. (**b**) Off-axis holography or reference-assisted holography technique can retrieve phase information from an interference pattern. However, it requires the use of a well-defined known reference beam. (**c**) Transmission matrix (TM)-based approach exploits the deterministic natures of light propagation through complex media, and enables the retrieval of phase information. However, it is generally based on off-axis holography and requires the use of a reference beam. (**d**) The SSM also exploits the deterministic nature of multiple light scattering. However, the phase information is exploited in terms of speckle-correlation, and removes the needs for a reference beam. Modified with permission from [36], NPG.

**Figure 2 sensors-20-03147-f002:**
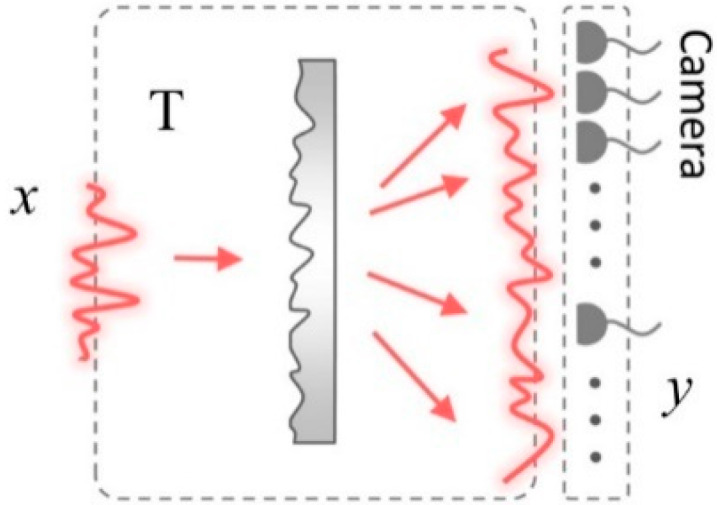
Schematic representation of the SSM method. An optical field incident **x** is scattered by an optical diffuser. The scattered field **y** is imaged by a camera. The scattered and the incident fields are related by the transmission matrix **T** of the optical system. Adapted with permission from [67], APS.

**Figure 3 sensors-20-03147-f003:**
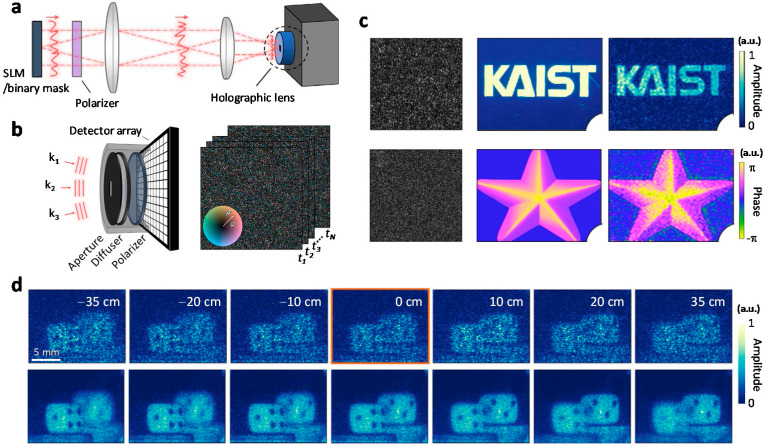
Reference-free holographic imaging using SSM. (**a**) Experimental setup for reference-free holographic imaging. SLM: spatial light modulator. (**b**) Schematic representation of the TM calibration process (left) and the transmission matrix. (**c**) Speckle images (left) measured by the camera for two different incident fields (middle). Reconstructed holographic images (right) form the speckle patterns. The upper and lower rows correspond to the result of amplitude-only and phase-only incident fields, respectively. (**d**) Measured holographic images of two dices (top) and compounded images obtained by changing the illumination (bottom). Images at different axial positions are obtained by numerical propagation. Adapted with permission from [36], CC-BY-4.0.

**Figure 4 sensors-20-03147-f004:**
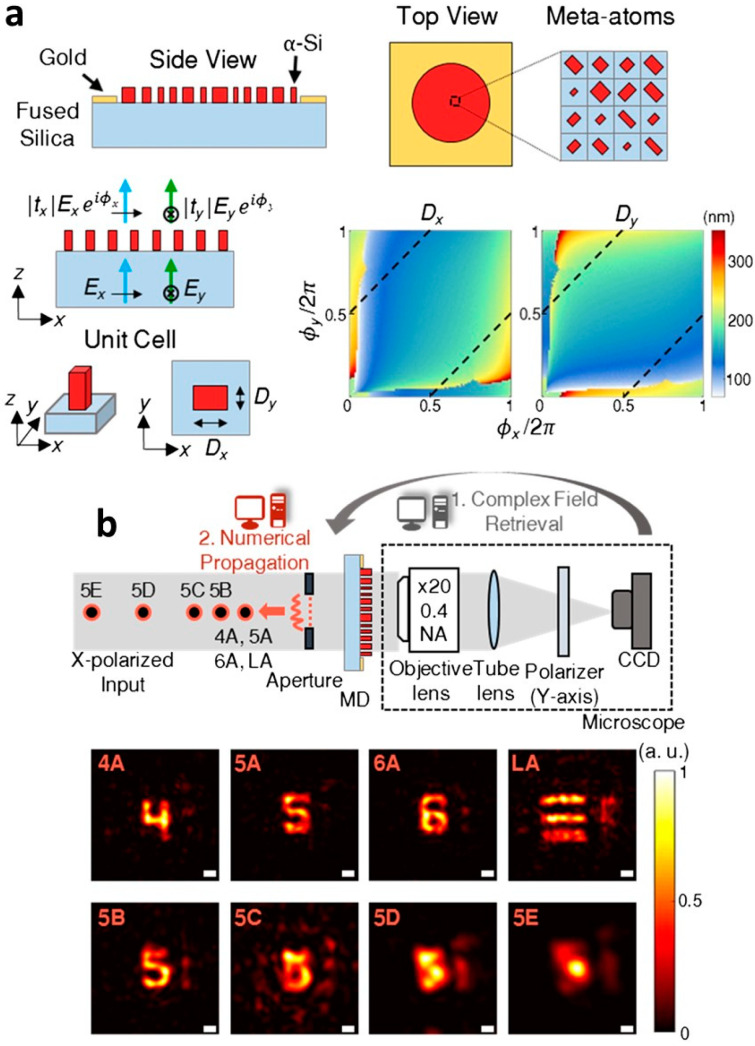
Reference-free holographic imaging using SSM. (**a**) The design of a metasurface diffuser for SSM. *A_x_* and *A_y_* are amplitude changes for x- and y- polarized light, respectively. *φ_x_* and *φ_y_* are induced phase delays for *x*- and *y*- polarized light, respectively. *D*_x_ and *D*_y_ represent the dimensions of the cuboid. The black dashed lines indicate the dimensions for each cuboid to act as a half-wave plate. (**b**) The measured holographic image of targets at different axial locations using the metasurface diffuser. The distances between the points and the aperture are as follows: A, 1.5 mm; B, 2 mm; C, 2.5 mm; D, 3.5 mm; and E, 4.5 mm. MD: metasurface diffuser, CCD: charge-coupled device. Scale bars are 25 μm. Adapted with permission from [69], OSA Publishing.

**Figure 5 sensors-20-03147-f005:**
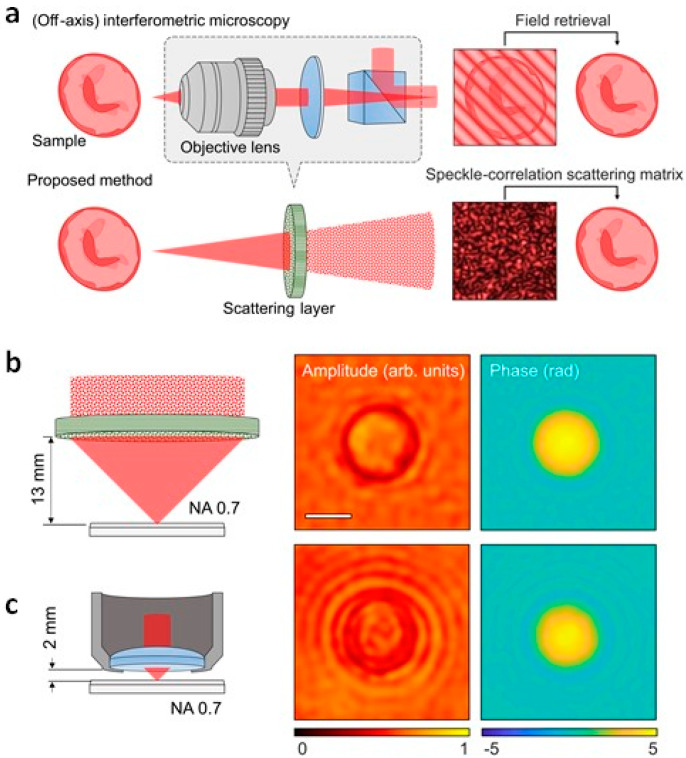
High-resolution long-working-distance holographic microscopy. (**a**) Schematic representation of holographic microscopy system based on conventional off-axis interferometry (top) and SSM (bottom). (**b**,**c**) The complex amplitude image of a 5 µm polystyrene bead measured using SSM-based microscopy (**b**) and a conventional method with a long-working distance objective lens (**c**). Adapted with permission from [51], APS. NA, numerical aperture.

**Figure 6 sensors-20-03147-f006:**
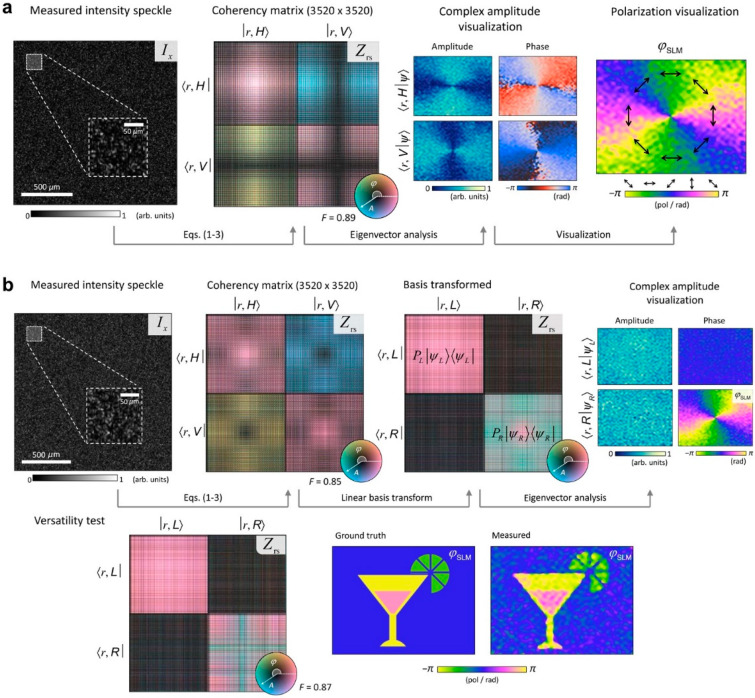
Coherency matrix and SSM. (**a**,**b**). SSMs measured from speckle patterns of polarized (**a**) and unpolarized light (**b**) become coherency matrices. *A* and *ϕ* are the amplitude and phase of light, respectively; *I_x_*_,_ is the measured intensity speckle in camera (*x*) domain; *Z_rs_* is the measured SSMs in the space–polarization (*r* and *s*, respectively) domain; *H*, *V*, *L*, and *R* are horizontal, vertical, left-circular, and right-circular polarizations, respectively; *P_L_* and *P_R_* are the statistical weight for the *L* and *R* polarizations, respectively; and $\langle r,L|\psi_{L}\rangle =\psi_{L}\left(r,L\right)$ and $\langle r,R|\psi_{R}\rangle =\psi_{R}\left(r,R\right)$ are the holographic images for the *L* and *R* polarizations, respectively. For both coherent and incoherent states of light, holographic images of two orthogonal polarizations are measured. Adapted with permission from [67], APS.

**Figure 7 sensors-20-03147-f007:**
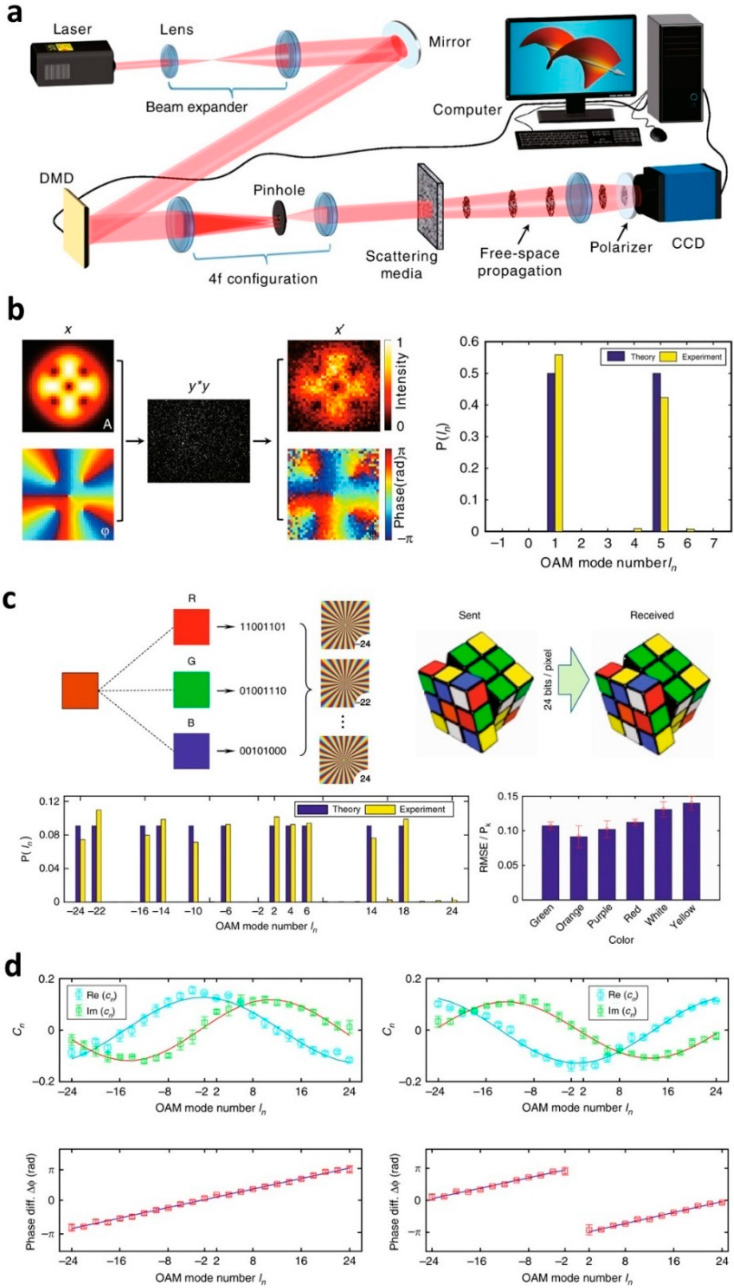
Orbital angular momentum (OAM) demultiplexing based on SSM. (**a**) The experimental setup for demultiplexing in a strongly scattering environment. (**b**) Demultiplexing of two OAM states. *x* is the incident field for two input modes, *y*^*^*y* is the measured speckle image, *x’* is the retrieved field using SSM, and P is the probability for OAM modes. (**c**) OAM-encoded data transfer through a scattering medium. RGB color information is encoded to deliver the image of a Rubik’s cube. Root mean square error (RMSE)/P_k_ is the normalized error of all colors in the received image of the Rubik’s cube, where P_k_ is the normalization coefficient defined as the inverse of the number of 1-value bits in a byte. (**d**) Phase modulation of 48 input OAM modes (top) and the measured relative phase (bottom). *c_n_* is the complex coefficient for the *n*-th OAM mode. Adapted with permission from [79], CC-BY-4.0.

**Figure 8 sensors-20-03147-f008:**
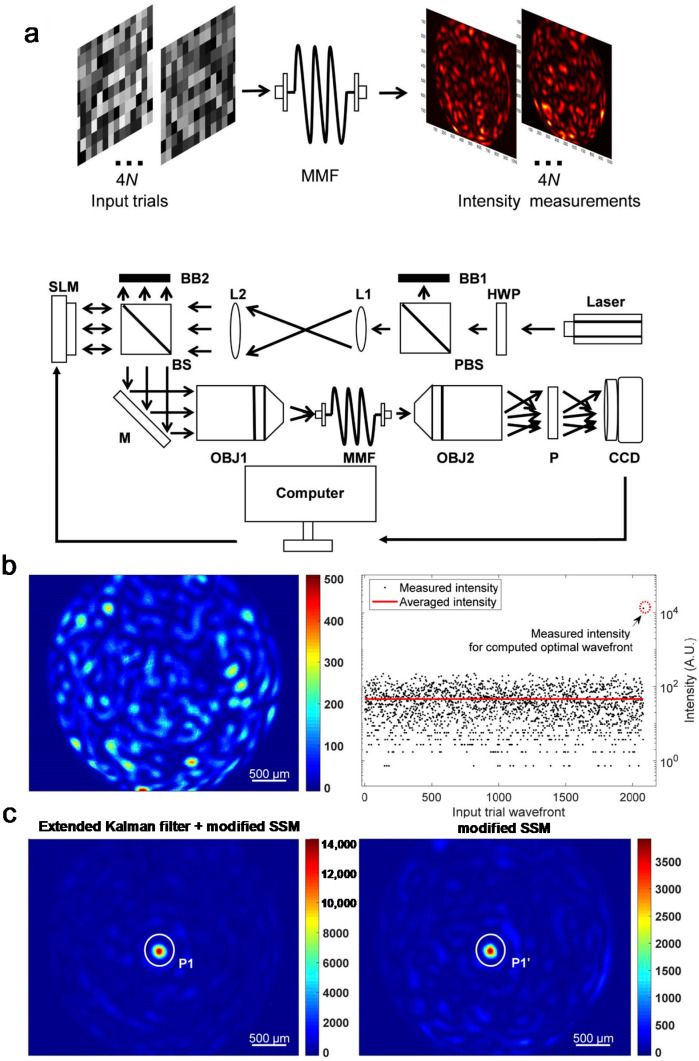
Calibrating the TM using modified SSM. (**a**) Schematic representation of the experimental setup. The input field is randomly modulated by an SLM and the corresponding fluctuation of the intensity is measured. BB: beam block; HWP: half wave plate; L: lens; M: mirror; MMF: multi-mode fibre; Obj: objective lens; P polarizer. (**b**) A speckle pattern transmitted through a multi-mode fibre (left) and the measured intensities at one output mode during the TM calibration process (right). (**c**) The wavefront shaping through the multi-mode fibre using the modified SSM with (left) and without (right) the extended Kalman filter. EKF: extended Kalman filter; MSSM: modified speckle-correlation scattering matrix. Adapted with permission from [52], OSA Publishing.

**Figure 9 sensors-20-03147-f009:**
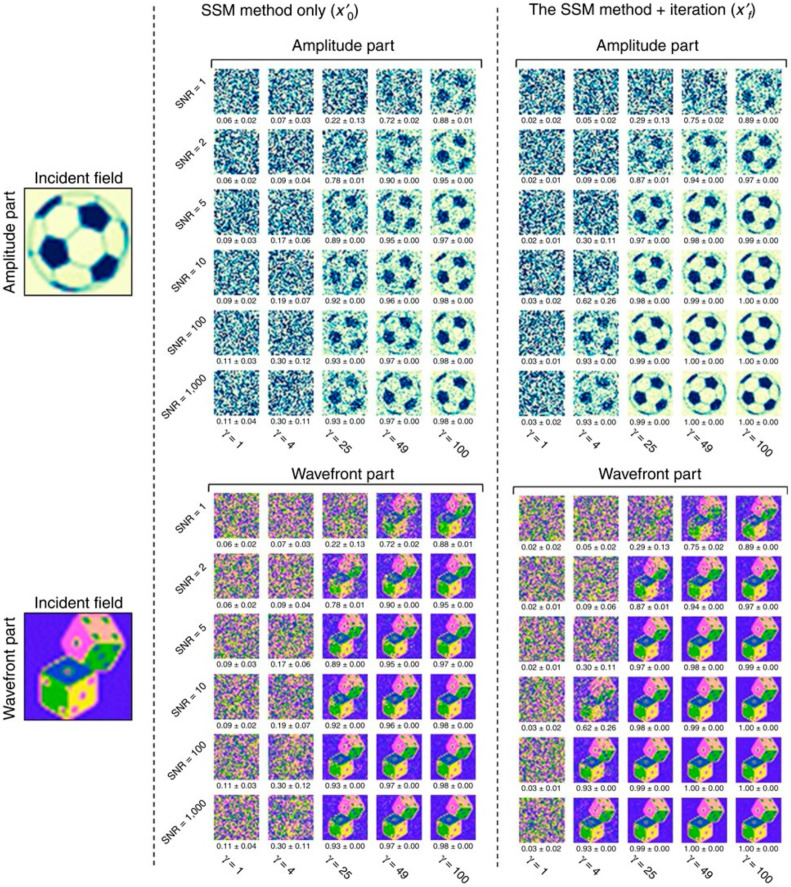
Amplitude and phase images before and after the error-reduction algorithm for the SSM with rank 1. Adapted from [36], CC-BY-4.0. SNR, signal-to-noise ratio.

**Figure 10 sensors-20-03147-f010:**
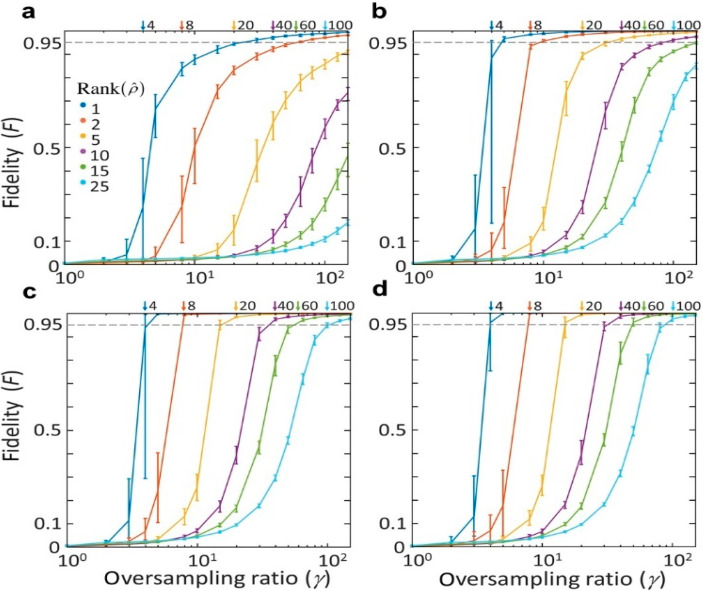
(**a**–**d**) The performance of the algorithm as a function of the rank of the SSM with SNR of 10 (**a**), 100 (**b**), 1000 (**c**), and infinity (**d**). The fidelity (*F*) is defined according to [92] and given by $F={\left[\mathrm{tr}\left(\sqrt{\sqrt{Z_{0}}Z\sqrt{Z_{0}}}\right)\right]}^{2}$, where Z_0_ and Z represent the prepared and measured states, respectively. Adapted with permission from [67], APS.

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
