# Peer review of "Speckle-Correlation Scattering Matrix Approaches for Imaging and Sensing through Turbidity"

_sensors, 2020, doi:10.3390/s20113147_

Round 1
Reviewer 1 Report
In this review, the authors present recent advances of speckle-correlation scattering matrix (SSM) techniques. The work gives some fundamentals of SSM approaches and details applications of the techniques in lensless imaging and sensing. The paper concludes on forthcoming SSM developments.
The paper would be of interest to readers of Sensors in the field of computational optics. After addressing some issues detailed below, it is my recommendation that the paper be accepted for publication in Sensors:
- Line 22 Keywords: “lens-less sensing” does make sense, maybe “sensing” is enough.
- Introduction section, line 44 maybe a comment must be added concerning deep learning implementation with lens-less imaging techniques considering for example the following reference: Yunzhe Li, Yujia Xue, and Lei Tian, "Deep speckle correlation: a deep learning approach toward scalable imaging through scattering media," Optica 5, 1181-1190 (2018). This might be also a way to improve the manuscript that only mentions deep learning as a future candidate in conclusion section, but works already exist in the field and should be presented in the proposed review to my point of view.
- Line 59: “Recent studies have demonstrated that the SSM methods serve as a universal approach…” In my opinion, the sentence is not clear enough and we do not understand what it refers to. Some references should be added here.
- Line 89: Fig. 1c not the right one, should be Fig. 1d
- Line 106: choose the following wording “square modulus” rather than “modulus square”.
- Maybe a discussion concerning the processing time can be added in the section “principle of SSM” giving, for example in sub-section 2.1, more information about the calibration process of the TM (Line 140), and also, about the holographic imaging process. Can we perform real-time imaging in the case?
- Section criteria for SSM: in the oversampling sub-section, line 350: controlling the size of the speckle grain, by adjusting distance scattering layer and sensor or size of the incident beam means that pixels of the camera maximize the oversampling required, but what about the dynamic of the pixel required? It permits also to improve the SNR, and errors. The following sub-section concerning iterative algorithms for error reduction does not explain the origin of the limited SNR, maybe a comment can help to clarify this aspect.
Reviewer 2 Report
Title: Speckle-correlation scattering matrix approaches for imaging and sensing through turbidity
This manuscript provides a general review on the emerging SSM technique from its principle to various applications and performance criteria. It is believed that this manuscript could be a good reference for the researchers in this field. I strongly recommend this manuscript for publication in MDPI Sensors.
[Minor correction]
Line 89, page 3
“ ~ a simple setup (Fig. 1c).” --> “ ~ a simple setup (Fig. 1d).”
Reviewer 3 Report
Manuscript the "Speckle-Correlation Scattering Matrix for imaging and sensing through turbidity" is devoted to a review of the developments of speckle-correlation scattering matrix (SSM) techniques, which is suitable for holographic imaging, microscopy and some other areas of optics.
The authors study a mathematical model of a unique technique known as the speckle correlation scattering matrix (SSM), which has several advantages over the TM technique and direct holographic technology.
It is a pity that the authors did not capture studies that were very close on the topic of recording information holograms without a reference beam, using a similar technique for reconstructing a hologram by those parts of interfering waves that do not carry an information component [Holographic Memory Without Reference Beam // doi: 10.3103 / S1060992X16040056 ].
As a remark, it should be noted that the authors of this good review paper were forced to very briefly describe the drawings presented. So, sometimes for their fuller understanding it was necessary to look at the original works from where these drawings were taken. For example, in the description of Figure 6 there are no notations at all both from the figure itself and from the formulas in the text that preceded the figure. This remark also applies to some other figures, such as, for example, Fig. 7,8
Undoubtedly, the technology considered by the authors will continue to actively develop and the work in question will be useful to many researchers. The work is very interesting and certainly should be published.
